# Optimal Scheduling to Manage an Electric Bus Fleet Overnight Charging

**Adnane Houbbadi [1],\* , Rochdi Trigui [1] , Serge Pelissier [1] , Eduardo Redondo-Iglesias [1] and Tanguy Bouton [2]**

1. Univ Lyon, IFSTTAR, AME, Eco7, F-69675 Lyon, France
2. TRANSDEV, 92130 Issy-les-Moulineaux, France
* Correspondence: adnane.houbbadi@ifsttar.fr; Tel.: +33-6-2039-9712

**Abstract:** Electro-mobility is increasing significantly in the urban public transport and continues to face important challenges. Electric bus fleets require high performance and extended longevity of lithium-ion battery at highly variable temperature and in different operating conditions. On the other hand, bus operators are more concerned about reducing operation and maintenance costs, which affects the battery aging cost and represents a significant economic parameter for the deployment of electric bus fleets. This paper introduces a methodological approach to manage overnight charging of an electric bus fleet. This approach identifies an optimal charging strategy that minimizes the battery aging cost (the cost of replacing the battery spread over the battery lifetime). The optimization constraints are related to the bus operating conditions, the electric vehicle supply equipment, and the power grid. The optimization evaluates the fitness function through the coupled modeling of electro-thermal and aging properties of lithium-ion batteries. Simulation results indicate a significant reduction in the battery capacity loss over 10 years of operation for the optimal charging strategy compared to three typical charging strategies.

**Keywords:** battery aging; electric buses; electric vehicles; electric vehicle supply equipment; nonlinear programming

## 1. Introduction

Electro-mobility in urban public transport is undergoing fast technological development and market expansion. Therefore, the use of electric buses (EBs) is a key aspect to address the challenges associated to urban transport for high energy efficiency and low greenhouse gas emissions. However, managing electro-mobility in local public transport is not an easy task. On the one hand, a large-scale deployment of EBs will have a great impact on the electrical system, generating peak-demand and network congestion problems [1]. On the other hand, the extra cost of EB purchase compared to thermal buses should be paid back in a reasonable time.

EBs operators are increasingly interested in reducing operation costs, particularly costs related to battery replacement. In order to reduce EBs battery aging, a smart use is needed so that the batteries operate in the most favorable conditions. Centralized and decentralized overnight charging scheduling for large and small-scale electric vehicle fleets has already been studied in order to minimize different objectives including V2G [2–12]. A centralized charging scheduling is operated by a central controller whereas decentralized charging scheduling is managed by individual users that optimize their own charging profiles. Some works have focused on the centralized overnight charging scheduling of EBs fleets [13–21]. Other works were dedicated to the optimization of charging infrastructure [22,23]. Some works have also addressed the battery aging issue in optimization by developing a capacity fade model for energy system that are not dedicated to transport [24,25].

The mentioned studies have investigated EBs fleets with overnight charging on centralized bus depots. Nevertheless, they analyze large and small-scale bus depots mostly from the perspective of minimizing the operational costs or the load peak without taking into account the aging of batteries.

The present paper differs from the referred ones [13–25] as it investigates more precisely the battery electro-thermal and aging behavior in order to minimize the battery aging cost. This work differs also from our previous results [26,27] that dealt with the reduction of the charging cost applied to a multi-objective small-scale EBs fleet and a mono-objective large-scale EBs fleet. The present study includes a large-scale EBs fleet with a possible extension of the proposed algorithm to hundreds of buses while using a mono-objective nonlinear programming optimization to minimize battery aging and requiring relatively short calculation time. To the best of our knowledge, there is no publication concerning a centralized overnight charging scheduling of large-scale EBs fleet taking into account battery aging.

This work addresses an optimal charging issue for EBs fleet by using centralized infrastructure to collect information about all EBs and centrally optimize EBs charging considering the grid technical constraints and real operating constraints of EBs.

The main contributions of the paper are summarized below:

- A methodological approach to manage an EBs fleet overnight charging based on nonlinear programing was carried out. The resulting smart charging strategy aims to minimize the battery aging cost by making optimal charging decisions. This approach can handle several operating constraints and the proposed algorithm could be extended to hundreds of buses with an acceptable computation time.
- An optimization tool was developed in the Matlab/Simulink environment. The tool optimizes a large-scale EBs fleet charging considering the grid technical constraints and real operating constraints. It includes an Electro-Thermal and Aging coupled battery model of a given EB to simulate the dynamic response of the battery.
- A case study was performed where the simulation of the proposed approach was conducted taking into account some real operating constraints. The potential economic gain of an optimal EBs charging for 10 years operation was compared to three typical non-optimal charging strategies to show the potential economic gain.

This paper is organized as follows: Section 2 introduces the charging infrastructure, the proposed approach to manage an EBs fleet overnight charging and the system modeling. Section 3 details the problem formulation and solving. Then, we present and discuss our results for a case study with different operating scenarios in Section 4. In Section 5, we draw some conclusions and introduce future work.

## 2. Methodology and System Modeling

### 2.1. Electric Vehicle Supply Equipment and Communication Protocol

An electric vehicle charging infrastructure, also called EVSE (Electric Vehicle Supply Equipment), is an infrastructure that provides electric energy, energy conversion, monitoring, and safety functionality according to international standards from the International Electrotechnical Commission. The charging type can be classified in two categories:

- The overnight charging where EBs batteries are charged at the depot overnight with slow chargers (typically 40–120 kW).
- The opportunity charging where EBs batteries are charged at bus stops (up to 600 kW) or terminals (usually between 150 to 500 kW) mainly using over-head pantographs.

Opportunity charging requires less energy to be stored in the bus, which could significantly reduces the capital costs; however, the cost of the required infrastructure remains very high compared

to the overnight charging infrastructure one. Both solutions are competitive in terms of total cost of ownership (TCO) and have their respective advantages and disadvantages [28]. In this paper, we focus on heavy vehicles (EBs) with overnight charging. Currently, charging mainly overnight is still possible for large-scale EBs fleet. However, in a case where the electricity demand exceeds supply, the charging should be rescheduled or modulated. The proposed optimization tool can handle this issue by imposing constraints from the power grid. The generalized system architecture is presented in Figure 1. The DC/DC charger is managed by a central control system. This controller checks the conversion equipment and carries out the communication with the EB to perform the charging according to a standardized communication protocol (PLC) EN 61851-23 [29]. A standard socket type Combo 2 CCS makes the hardware connection between the EB and the charger. Further details about the system operation, communication and charging protocol are explained in [30,31].

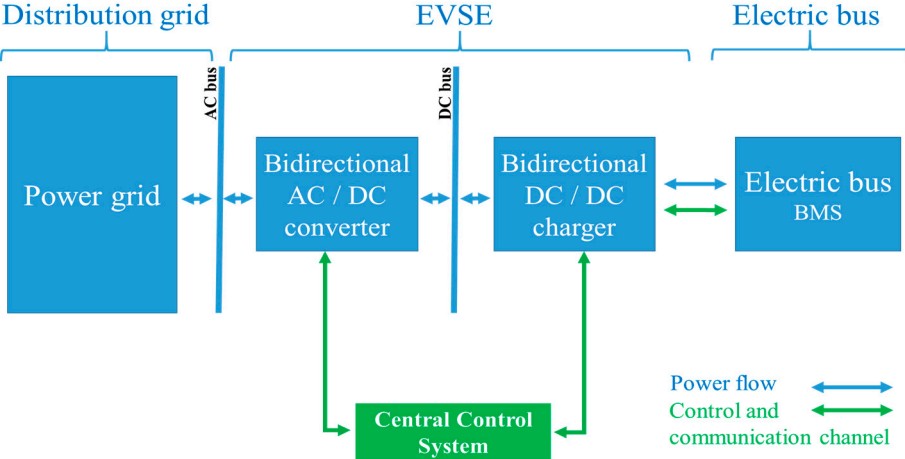

**Figure 1.** System architecture of the EVSE.

### 2.2. Optimization Tool for the Management of the EBs Fleet Charging

This methodological part describes the approach used to optimize a charging strategy for a fleet of EBs charging overnight as shown in Figure 2. This approach uses nonlinear programming to search for optimality according to the objective function (which is here to minimize the battery aging cost) while taking into account various constraints (actual operating constraints, power grid and the charging station constraints). The optimization methodology has been executed in the Matlab/Simulink environment. Before optimizing the charging process, information is required regarding the number of buses and the operating constraints of each EB: initial state of charge (SoC), initial battery temperature, the target SoC, arrival and departure time, maintenance period.

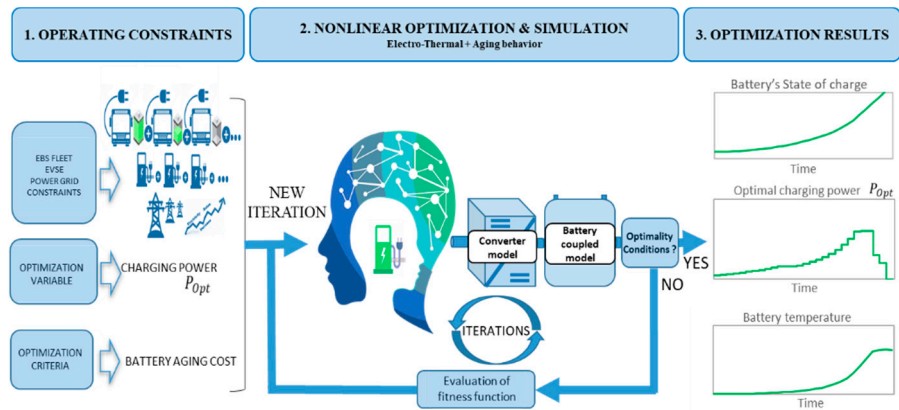

**Figure 2.** Smart charging methodology for the management of EBs fleet charging.

Other constraints related to the charging infrastructure and power grid have to be considered. For instance, the subscribed power or the maximum charging power delivered by the charger. The optimization algorithm attributes an optimal charging power for each bus depending on objectives and respecting all the constraints.

### 2.3. Electro-Thermal and Aging Coupled Model

#### 2.3.1. Battery Electrical Model

Several equivalent electrical models for Li-ion batteries are proposed by the literature [32]. The simplest model to describe the behavior of an accumulator is the OCV+R model, which associates a voltage source and a resistance. For the purpose of this work, this model provides the necessary accuracy with a very short computation time.

This model was previously developed in the VEHLIB (Vehicle Electrique Hybride LIBrary) project [33]. This software created by IFSTTAR is a systemic tool for evaluating the energetic and dynamic performances of conventional, electric and hybrid vehicles.

The equivalent electric circuit equation is given by:

$$U_{bat} = OCV - R_{eq} \times I_{bat} \tag{1}$$

This model implements an ideal voltage source $OCV$ that represents the open circuit voltage of the battery, and a resistor $R_{eq}$ that includes an ohmic, double layer and diffusion resistance. Resistance and $OCV$ are a function of SoC and temperature. $I_{bat}$ is the battery current with a positive value when discharging and a negative value when charging. $U_{bat}$ is the battery voltage. Battery electrical characteristics are presented in Table 1, Section 4.

**Table 1.** Simulation parameters input, operating constraints and electro-thermal-aging battery characteristics.

| Parameters | Value | Parameters | Value |
|---|---|---|---|
| Number of buses | 1 to 10 | Battery type | LIFePO4 or LFP |
| Number of simulated days | 1 day | Nominal energy/capacity | 311 kWh/540 Ah |
| Charging time period | 13 h 30 | Pack surface for thermal exchange | 18.79 m$^2$ |
| Charging slot $\Delta t$ | 30 min | Battery pack weight | 2500 kg |
| Charging power for 1 charger | $P_{max}$ | Specific heat capacity | 900 J·kg$^{-1}$·K$^{-1}$ |
| Number of charging time slots | 27 | Heat transfer coefficient | 5 W·m$^{-2}$·K$^{-1}$ |
| Initial state of charge | 10% | A: pre-exponential factor | $4.35 \times 10^7$ p.u.day$^{-1}$ |
| Target state of charge | 100% | E$_a$: activation energy | 0.719 eV |
| Arrival & departure time | $t_0 \rightarrow t_0 + 14$ h | k: Boltzmann constant | $8.617 \times 10^{-5}$ eV·K$^{-1}$ |
| Initial battery temperature | 25 °C | B: quantity of charge factor | 1.104 |
| Initial battery capacity fade | 0% | Q$_{EoL}$: Capacity loss at end of life | 0.2 p.u |
| Fixed outside temperature | 25 °C | Bat$_{price}$: Battery price | 500 €·kWh$^{-1}$ |

#### 2.3.2. Battery Thermal Model

Different thermal modeling approaches are given by the literature. The widely used one is based on the finite element model. There are also other approaches using artificial neural network or partial differential equation model [34]. The advantage of these models is their high accuracy. However, these approaches are complex to implement and require a long computation time.

To overcome these difficulties, thermal models are simplified to the form of equivalent thermal circuits (ETC), where resistors, capacitors, and current sources are used for representing heat transfer, heat accumulation and heat source, respectively. In this work, it is not useful to simulate very finely the different internal temperature gradients in the battery. ETC based models represent a good compromise between the complexity and accuracy of our system.

The battery thermal model used is a simple model of a prismatic battery (LiFePO4/graphite) based on an equivalent electrical circuit [35]. We supposed that the cell temperatures inside the battery pack

are homogeneous and that the temperature is identical at the core and at terminals of each battery cell. The studied one-node model shown in Figure 3 contains a quantity Q of heat that is generated by the current source, one-capacity $C_{th}$ that represents heat accumulation and a thermal resistance $R_{th}$ that represents convection heat transfers with ambient air. The cooling system is not considered for the moment.

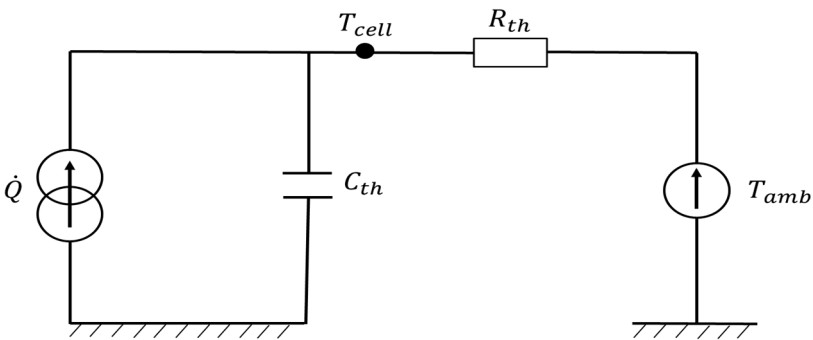

**Figure 3.** Battery thermal model.

According to [36], heat generated by battery during its working process could be divided into three parts: the reversible entropic heat $Q_r$, side reaction heat $Q_s$, and Joule heat $Q_j$. Actually, $Q_s$ is small enough to be neglected. The total heat $Q$ generated by the battery during its working process could be simplified as Equation (2) shows.

$$Q = Q_r + Q_j \tag{2}$$

In the reference [37], $Q_r$ is only approximately 6–7% of the total heat generated. For large batteries used on EVs, the Joule heat mainly dominates heat generation.

$$Q \approx Q_j = I_{bat}\,(U_{bat} - OCV) \tag{3}$$

In addition to heat generation, the battery would also dissipate heat to the ambient since the battery pack is not adiabatic. We assumed that conduction and heat radiation losses could be neglected. The dissipated heat $Q_{dis}$ is expressed as follows:

$$Q_{dis} = h\,(T_{bat} - T_{amb}) \tag{4}$$

where $h$ is the heat transfer coefficient with $h = \frac{1}{R_{th}}$, $T_{bat}$ is the battery temperature, $T_{amb}$ is the ambient temperature. In combination of (3) and (4), the total heat $Q_T$ can be described as shown in Equation (5):

$$Q_T = Q - Q_{dis} \tag{5}$$

The temperature change of the battery is the effect of heat generated by the battery and the heat dissipated from the battery.

$$C_{th}\frac{dT_{bat}}{dt} = I_{bat}\,(U_{bat} - OCV) - h\,(T_{bat} - T_{amb}) \tag{6}$$

where $I_{bat}$ is the battery charge current, $U_{bat}$ is the battery voltage, $OCV$ is the open circuit voltage, $C_{th}$ is the specific thermal capacity of the battery. Battery thermal characteristics (heat transfer coefficient, thermal capacity) are presented in Table 1, Section 4.

### 2.3.3. Battery Aging Model

The battery performance (power, capacity) decays over time due to multiple aging mechanisms. These mechanisms are side reactions and depend on the battery conditions (T, SoC) and battery solicitation (profile of current, cumulative of A.h). Battery aging can be categorized into calendar (I = 0) and cycling aging (I ≠ 0).

Calendar aging includes all aging processes that lead to a degradation of a battery cell independent of charge-discharge cycling. In recent lithium-ion batteries, the main aging process contributing to calendar aging is the Solid Electrolyte Interphase (SEI) formation on the negative electrode [38]. Depending on the ambient temperature and SoC of the battery, the calendar aging could be more or less important. It is an important factor in many applications of lithium-ion batteries such as in EVs. Furthermore, the degradation due to calendar aging can also be predominant in cycle aging studies, especially when cycle depths and current rates are low.

Cycling aging occurs when using the battery. The aging severity depends on current rates, SoC and battery temperature [39].

In this work, it must be highlighted that cycling aging could be neglected over the period of slow charging as current is very low (C-rate ≤ C/6) and the ambient temperature is set to 25 °C [40]. As for battery calendar aging evaluation, we used an empirical model proposed by Redondo-Iglesias [38] to evaluate infinitesimal variation of calendar capacity loss. The formula of this semi-empirical model is based on the Eyring equation:

$$\dot{Q}_{loss} = A \cdot \exp\left(-\frac{E_a}{k\,T} + B \cdot Q_a\right) \tag{8}$$

where, $\dot{Q}_{loss}$ is the capacity loss rate (p.u./day), $A$ is the pre-exponential factor (p.u./day), $E_a$ represents the activation energy for the reaction (eV), $k$ the Boltzmann constant (eV/K), $T$ the absolute temperature (K), $B$ the quantity of charge factor (no units), $Q_a$ the available quantity of charge (p.u.). All the aging parameters values are presented in Table 1 from [38].

### 2.3.4. Electro-Thermal and Aging Coupled Battery Model

To make the entire battery model more accurate, the three aforementioned sub-models are interconnected. Battery equivalent resistance and voltage are temperature dependent. Therefore, the electrical model needs to communicate with the thermal model. On the other hand, battery aging leads to capacity loss. This degradation has to be evaluated by the aging model and referred back to the electrical model so that the SoC evaluation takes into account the actual battery capacity.

The battery coupled model works as follows: the model receives a given converter power input and initial values of SoC, battery temperature and aging state. The electrical model computes the dynamic behavior of the battery during a specified duration and provides various values such as the SoC variation, the equivalent resistance, battery current and voltage. The thermal model provides the dynamic behavior of the battery temperature within the specified period. This model employs information like the equivalent resistance and the battery current to provide the battery temperature variation. The aging model determines the battery capacity fade. This model uses data of the SoC variation and the battery temperature variation.

Once the period is over, the variation of SoC, temperature and capacity fade are added to the initial input values. This process is repeated for the next period of simulation. Figure 4 shows the coupled battery model.

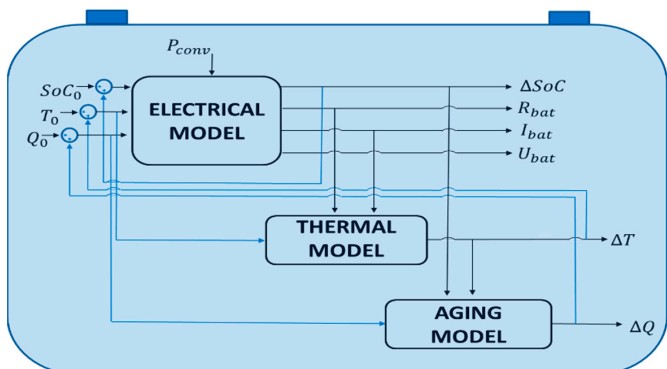

**Figure 4.** Battery coupled model.

### 2.3.5. Converter Model

To represent the charger efficiency, a converter model is developed and added upstream of the battery coupled model as shown in Figure 5. The efficiency of an electric vehicle (EV) charger depends on the efficiency of various internal components. Electrical power comes from the grid and is converted several times, from the AC supply to the AC/DC converter and from the DC/DC converter to the battery. In order to stay in an acceptable simulation time, we used some experimental results of charger test efficiency for our converter model to represent the AC/DC and the DC/DC converter power losses [41]. These data give the average charger efficiency during the charging tests for different charging powers.

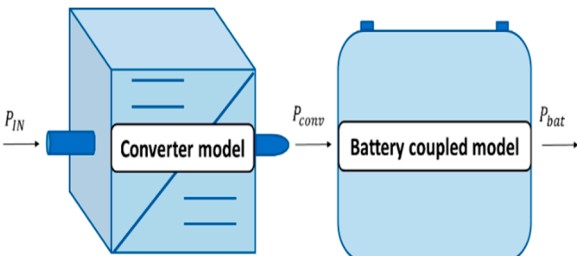

**Figure 5.** Converter and battery coupled model.

In a first approach, before developing a specific converter model, we used a linear interpolation of charger efficiency values based on the experimental results mentioned above.

## 3. Optimization Problem Formulation

The references [2–12] studied the centralized and decentralized overnight charging scheduling of large and small-scale EV fleet in order to minimize different objectives including V2G. The references [2–4,8] investigated the battery life prolongation through an optimized use of the charging/discharging process of the battery by minimizing the operational cost. References [5,6] managed a large number of EVs in order to minimize cost with respect to special grid constraints. References [7,9,10] coordinated the charging of EVs to flatten the loading profile of the transformer substation, to reduce peak demand and to minimize costs while respecting EV users and technical constraints. In this work, we focus on the deployment of large-scale heavy vehicles (EBs).

### 3.1. Literature Review of Large-Scale EBs Smart Charging Algorithms

Several methods for smart charging of large-scale electric vehicles were compared through an extensive literature review [12]. Hu et al. have compared nine optimization methods used for smart charging of EVs with different objectives (minimizing charging cost, power losses, load levelling, network voltage deviations and providing regulation services). A linear programming based technique is highly recommended for a large-scale EVs fleet because it is the fastest one and highly efficient for finding the optimal solution.

However, in terms of applicability, this method is not suitable for non-linear problems for instance, the battery aging issue. Some other optimization methods that deal with non-linear problems (non-linear programming, heuristic and meta-heuristic optimization) were also discussed and compared.

Regarding the charging scheduling of large-scale EBs fleets:

Rinaldi et al. [13] used a mixed integer linear programming to minimize the charging costs for a bus fleet of up to 8 buses. Gao et al. [14] developed a consumption prediction model and an optimization charging model of large-scale EBs fleet. They used genetic algorithms to minimize the EBs operating cost in four different bus lines in China. Jahic et al. [15] analyzed the charging scheduling problem on 40 EBs and proposed two algorithms (Greedy and heuristic algorithm) for the load peak minimization. Guyot et al. [16] presented an overnight bus charging strategy at the bus depot of 114 buses in order to minimize the peak load. Jiang et al. [17] recommended a neighborhood search based heuristic for

scheduling large-scale EBs fleet on a real bus route in Shenzhen, China. The objective was to minimize the annualized total cost. Leou et al. [18] proposed a stochastic approach for minimizing energy costs of 10 EBs on a centralized depot. The model took into account the variable electricity prices during the day. Chen et al. [19] proposed two real-time coordinated charging strategies of EBs fleet to minimize both the charging costs and the peak load. They used a mixed integer linear programming at first, then, to reduce the computation time, they used a suboptimal strategy with a heuristic method. Wang et al. [20] analyzed a real-world dataset of 16,359 EBs, 1400 bus lines and 5562 bus stops, which is obtained from the Chinese city Shenzhen. They designed "bCharge", which is a real-time charging scheduling system. The charging scheduling problem was formulated as a Markov Decision Process (MDP) problem to reduce the charging cost for large-scale EBs fleet. T. Zhu et al. [21] optimized the preheating of the Li-ion battery of an EB in order to reduce battery aging. This study used a coupled electro-thermal-aging model during the traction use phase.

While most existing works focus on minimizing the operational costs or the peak load, to our best knowledge, there are no research works analyzing the battery aging issue with a large-scale EBs fleet. In this paper, regarding the nonlinear form of the battery aging function and the nonlinear constraints, we will focus on algorithms for solving nonlinear function.

Nonlinear programming methods over heuristic and meta-heuristic algorithms work remarkably well for a great majority of big-sized problems without too many local minima. The nonlinear programming quickly reaches a local solution but does not explore a larger space. Meta-heuristic requires more function evaluations, and searches through several basins to find a better solution. However, they are generally time consuming and more suitable to non-derivative-based optimization problem with multiple local optima. Considering our large-scale (EBs fleet) optimization problem with a large number of parameters, the nonlinear form of the objective function and the function differentiability, nonlinear programming has been selected with the expectation of a fast calculation time and sufficiently accurate results.

### 3.2. Nonlinear Programming Optimization (NLP)

NLP is the process of solving an optimization problem where objective function and some of the constraints are nonlinear. NLP uses different methods to solve optimization problems. The most common one is the sequential quadratic programing, which is a gradient-based optimization method where the gradient of the objective function and a linear step are calculated at each iteration, in order to update the variables. Other methods could be used such as the interior point method or the trust region method [42].

### 3.3. Pre-Optimization Process

Before applying the optimization process, we need to determine the amount of energy required to reach the targeted SoC of the battery. We used the battery-coupled model with different charging power and different SoC values to calculate the amount of energy required to reach the final SoC.

### 3.4. Optimization Design Variable

In this work, the optimization variable represents the charging power $P$ of the bus fleet in an (n × m) matrix:

$$P = \begin{pmatrix} p_{1,1} & p_{1,2} & \cdots & p_{1,m} \\ \ldots & \ldots & \ddots & \ldots \\ p_{n,1} & p_{n,2} & \cdots & p_{n,m} \end{pmatrix} = \left( p_{i,j} \right) \mid \left\{ \begin{array}{l} i = 1, 2 \ldots, n \\ j = 1, 2, \ldots, m \end{array} \right\} \tag{9}$$

where $n$ is the total number of EBs and $m$ the number of time slots, $p_{i,j}$ is the charging power of a bus number $i$ during a time slot $j$.

### 3.5. Objective Function

The aim is to minimize the battery aging cost. According to automotive standards, a 20% capacity loss generally indicates the battery end of life (EOL) [21]. This problem can be formulated as follows:

$$\min \sum_{i=1}^{n} \sum_{j=1}^{m} \frac{\Delta Q_{lossi,j}}{Q_{EoL}} \times Bat_{price} \times E_{bat\ i,j} \tag{10}$$

The battery aging cost represents the cost of replacing the battery spread over the battery lifetime.

This cost is expressed as a nonlinear function where $\Delta Q_{lossi,j}$ is the capacity loss rate of a bus number $i$ during a time slot $j$ (p.u./day), $Q_{EoL}$ is the capacity loss at end of life (p.u.), $Bat_{price}$ is the price of the LiFePO4 battery (€/kWh) and $E_{bat\ i,j}$ is the amount of energy of a bus number $i$ required during a time slot $j$ (kWh).

### 3.6. Linear Equality and Inequality Constraints

During optimization, the maximum charging power $p_{max}$ delivered by the charging infrastructure imposes a bound constraint to all elements of the optimization variable $P$.

$$0 \leq p_{i,j} \leq p_{max} \quad \left\{ \begin{array}{l} i = 1, 2 \ldots, n \\ j = 1, 2, \ldots, m \end{array} \right\} \tag{11}$$

The charging infrastructure subscribed power $p_{subscribed}$ imposes an inequality constraint to the submatrix $P_{\{j\}}$.

$$A \times P_{\{j\}} \leq p_{subscribed} \tag{12}$$

A denotes an all-ones (1 × n) matrix, $P_{\{j\}}$ is an (n × 1) refers to the charging power for all the buses in a time slot $j$.

The operating bus constraints (the bus arrival and departure time) impose an equality constraint to the submatrix $P_{\{i\}}$.

$$B \times P_{\{i\}}{}^{T} = 0 \tag{13}$$

B denotes an all-ones (1 × m) matrix that will take 0 between the bus arrival and departure time, $P_{\{i\}}{}^{T}$ (the transpose of the matrix $P_{\{i\}}$) is an (m × 1) matrix that represents the charging power of a bus number $i$ during the total time slots.

The operating bus constraint (number of km to be covered during the following day) will define the final battery SoC to reach. It also imposes the amount of energy required to reach the targeted SoC as an equality constraint to the submatrix $P_{\{i\}}$.

$$C \times P_{\{i\}}{}^{T} = \frac{\left(SoC_{final(i)} - SoC_{initial(i)}\right) \times Q_{bat(i)} \times V_{bat(i)}}{100 \times \Delta T \times \eta_{ch(i)} \times \eta_{bat(i)}} \tag{14}$$

C denotes an all-ones (1 × m) matrix, $P_{\{i\}}{}^{T}$ is an (m × 1) matrix that represents the total charging power of a bus number $i$ (W), $\Delta T$ the time-slot of 30 min (h), and $Q_{bat}$ denotes the total battery capacity of a bus number $i$ (Ah), ηch is the average charger efficiency of a bus number $i$ (%), ηbat is the average battery efficiency of a bus number $i$ (%), $V_{bat}$ the average battery voltage of a bus number $i$ (V). The average values are calculated during the pre-optimization process.

### 3.7. Nonlinear Equality and Inequality Constraints

The charging process of a Li-ion battery contains a constant current phase (CC) and a final constant voltage (CV) phase [43]. The end of the charging process has also to be carefully modeled. Considering the slow charging rate used in the case studied in this work, we assumed that the CV phase lasts one hour (2 slots of 30 min). As the CV phase is used to limit the current, we set the power to two decreasing

power values ($p_1$, $p_2$) during a time slot, in such a way that the CV phase charging allows the SoC to increase from 95% to 100%. This charging limitation can be expressed as a nonlinear equality:

$$p_{i,j} = p_1 \tag{15}$$

$$p_{i,j+1} = p_2 \tag{16}$$

where $p_{i,j}$ is the power value during the penultimate time slot, $p_{i,j+1}$ is the power value during the last time slot.

Code 1 represents the pseudocode of the optimization steps described in Section 3.

---

**Code 1** Proposed optimization steps

---

1- Initialize the number of buses $Eb = 1 : n$

2- Define the linear (in)equalities and lower/upper bounds: *lb, ub, A, B, C*

$lb = 0 \mid ub = p_{max};\ A \times P_{\{j\}} \le p_{subscribed};\ B \times P_{\{i\}}{}^T = 0$

$C \times P_{\{i\}}{}^T = \frac{\left(SoC_{final(i)} - SoC_{initial(i)}\right) \times Q_{bat(i)} \times V_{bat(i)}}{100 \times \Delta T \times \eta_{ch(i)} \times \eta_{bat(i)}}$ *(Pre-optimization process)*

3- Define the nonlinear equalities: $Ceq(1) = p_{i,j+1} - p_1\ Ceq(2) = p_{i,j} - p_2$

4- Define randomly an initial point $P_0$ (n×m) matrix

$$P_0 = \begin{pmatrix} p_{1,1} & p_{1,2} & \cdots & p_{1,m} \\ \ldots & \ldots & \ddots & \ldots \\ p_{n,1} & p_{n,2} & \cdots & p_{n,m} \end{pmatrix}$$

5- Define the objective function fun $= \sum\limits_{i=1}^{n} \sum\limits_{j=1}^{m} \frac{\Delta Q_{loss\,i,j}}{Q_{EoL}} \times Bat_{price} \times E_{bat\,i,j}$

6- Optimization process

    **for** i = 1: MaxIter

        Calculate Gradient around current point $P_i$

        Generate the next solution $P_{i+1}$

        Evaluate the next solution fun $(P_{i+1})$

        Report the optimum solution

    **end**

7- Optimization results: $P_{opt}$

An optimal charging power depending on objectives and respecting all the constraints

---

## 4. Case Study

A case study was performed to illustrate the optimization of an EBs fleet. The considered station is capable of supporting the simultaneous recharge of 10 EBs. The schematic of the setup is shown in Figure 6. The EBs will run during the day on an existing conventional bus line. The optimization takes place during the overnight charging when the EBs return to the depot. A total of 10 DC/DC converters power 10 chargers with standard Combo CC2 plugs.

The EBs fleet is charged only at the depot during a period of 13.5 h divided into 27 time-slots of 30 min each, which results in a reasonable search space for the optimization. Each bus of the EBs fleet arrives to the depot at a different time with different initial SoC depending on real operating constraints. We fixed the battery initial temperature and the ambient temperature to 25 °C, the initial capacity fade was fixed to zero.

The data summarized in Table 1 correspond to the simulation parameters of our optimization model for the above mentioned case study. The electro-thermal and aging characteristics of the battery pack are also presented in Table 1.

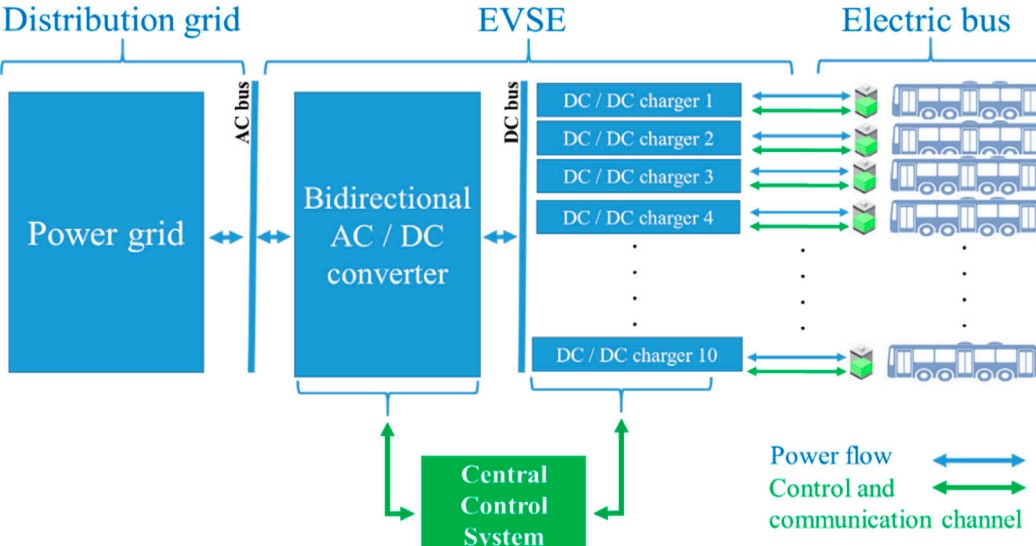

**Figure 6.** Charging infrastructure.

The battery price was set at 500 €·kWh$^{-1}$. The reference [44] proposed a Li-ion battery capital cost ranges between 500–2500 \$·kWh$^{-1}$. Other research [45] proposed a Li-ion battery pack cost for EV ranges between 250–360 \$·kWh$^{-1}$ for 2015, 200–250 \$·kWh$^{-1}$ for 2020 and 150–200 \$·kWh$^{-1}$ for 2025. The cost estimation uncertainty of lithium-ion cells depends on several factors. Different values were found in the literature, thus, a mean value of 500 €·kWh$^{-1}$ was chosen in this study as a realistic example (which seems more acceptable).

To estimate the heat capacity and heat transfer coefficients, experiments were conducted on a Citroen C-Zero battery pack. The vehicle was stored for a long time under ambient temperature (20 °C) to measure the heat transfer coefficient, and then fully charged at a current rate of C/7 to measure the corresponding heat capacity. We found a Cp = 900 J/kg/K and h = 5 W/m$^2$/K. These values were compared with experimental values from the literature. The reference [21] used a Cp value of 854 J/kg/K given by the battery manufacturer for a lithium iron phosphate battery (LIFePO4 or LFP). In the same study, the average value of 15.86 W/m$^2$/K was used for h including a thermal insulation. The battery thermal characteristics are strongly dependent on the battery pack configuration. We assumed that we have similar configuration as the Citroen C-Zero battery pack with no thermal insulation, no forced air convection and no cooling system.

## 5. Results and Discussions

In this study, we carried out mono-objective optimization by minimizing the battery aging cost regardless of the electricity cost. As a first step, we thus presented an optimal solution to minimize the aging cost for one EB, then, for two EBs, subjected to strong constraints and finally we discussed an operation cost review comparison between the optimal solution and three typical charging strategies. The optimization algorithm managed a larger size of bus fleets (up to 10 EBs) with several constraints during a short computing time. Here, we focus on presenting the results of a small fleet to carefully understand and analyze the algorithm behavior.

### 5.1. Optimization of the Aging Cost for One EB Charging

The first scenario concerns the case of one EB that arrives at $t_0$ with an initial SoC of 10% and must be fully charged at $t_0$+ 14 h.

The results of this mono-objective optimization for minimizing the battery aging cost presented in Figure 7 show that the optimal charging power tends to increase gradually. This seems logical according to the aging fitness function defined in Equation (8). The optimal charging power profile

chosen by the optimization algorithm leads to a delayed charge with lower values of SoC and charging power to minimize calendar aging and battery overheating respectively. The last two power slots values are dedicated to the end of charge as mentioned in Section 3.7. The "Greedy" baseline strategy typically used to charge the EB with the maximum power as soon as possible causes a faster temperature rise as we can see in Figure 7.

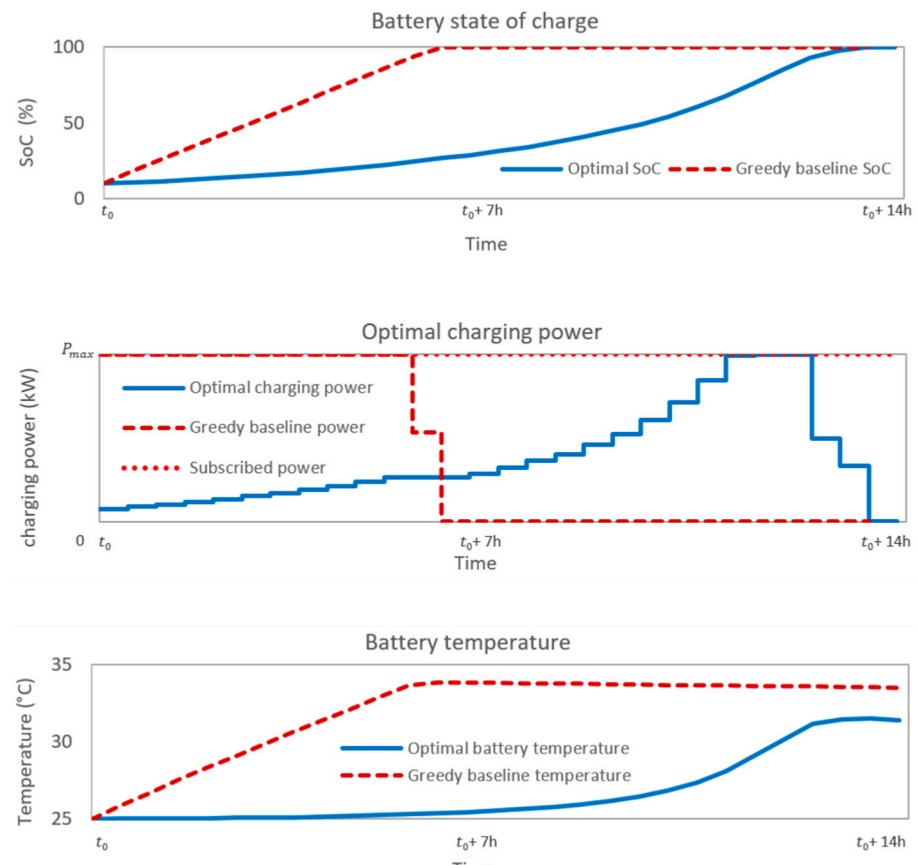

**Figure 7.** Optimal charging schedules scenario 1 and greedy baseline strategy.

### 5.2. Optimization of the Aging Cost for Two EBs Charging

In scenario No. 2, we performed a charging optimization for a fleet of two buses by adding some real operating constraints to see if the optimization algorithm will converge while respecting all the operating constraints. The first EB arrives at $t_0$ with an initial SoC of 10% and must be fully charged at $t_0 + 11$ h. The second EB arrives with an initial SoC of 10% at $t_0 + 5$ h and must be fully charged at $t_0 + 14$ h. The results of this mono-objective optimization for minimizing battery aging cost are presented in Figure 8 and show that the optimal charging strategy is to power the two EBs **EB1** and **EB2** during all the possible charging periods (when each bus is available). The algorithm ensures that the EBs charging is done progressively without exceeding the subscribed power. At $t_0 + 5$ h, the algorithm will manage the **EB2** arrival by decreasing the charging power of **EB1**. As a result of several constraints, the algorithm was forced to decreasingly charge **EB2**, otherwise the power would exceed the subscribed power.

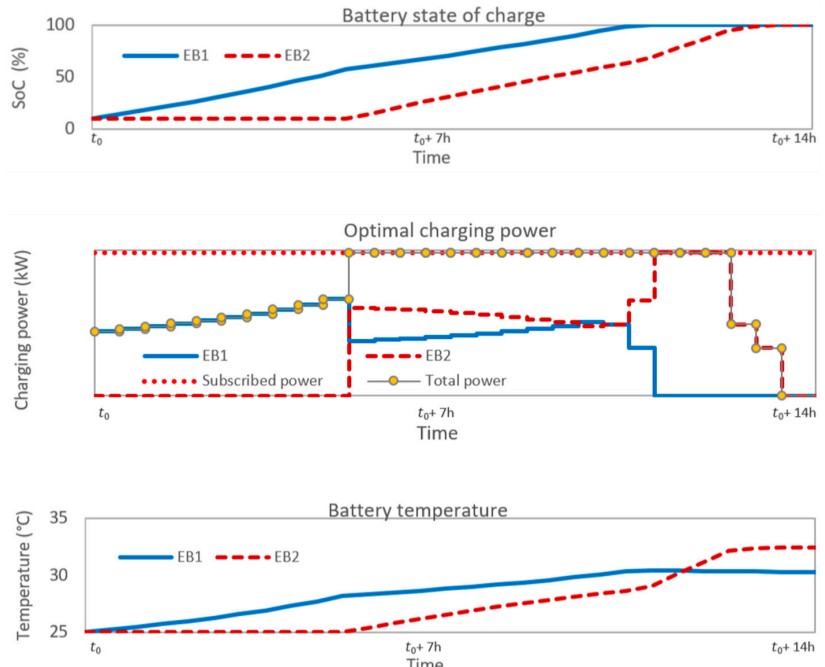

**Figure 8.** Optimal charging schedules scenario 2.

Table 2 presents the CPU (Central Processing Unit) time for the charging optimization of different EBs fleet sizes over a period of 1 day. The NLP programming was compared with non-dominated sorting genetic algorithm (NSGA-II) used in a previous work [26]. From a bus fleet larger than 10 EBs and more, NSGA-II does not converge.

**Table 2.** CPU (Central Processing Unit) time.

| EBs Fleet | 1 EB | 4 EBs | 10 EBs | 50 EBs | 100 EBs |
|-----------|------|-------|--------|--------|---------|
| NLP | 1–10 s | 10–15 s | 20 s–1 min | 5–15 min | 5–15 min |
| NSGA-II | 2 min | 20 min | 1 h | No convergence | No convergence |

### 5.3. Baseline Comparison and Annual Cost Review

To illustrate the potential economic savings, we will optimize the battery aging cost of one EB for 10 years of operation. We will use the same operating constraints as in the scenario no. 1.

The optimized charging strategy will be compared to three typical charging strategies:

- The "Greedy" baseline represents one typical behavior where the EB is charged with the maximum power as soon as possible, ignoring charging cost, until it is fully charged.
- The "Medium" baseline represents one typical behavior where the EB is charged with an average power during the full charging time.
- The "Postponed" baseline represents one typical behavior where the charging of the EB is postponed as late as possible.

Other charging strategies have been compared in the literature [7,46]. In this work, we compared our strategy's results to the three typical strategies mentioned before, as they are easy to implement.

The results of this mono-objective optimization for minimizing battery aging cost in Figure 9 show that the optimal charging strategy is not necessarily the one that stayed on lower value of SoC nor the one that minimizes the temperature. The optimal charging strategy achieves a good balance between the effects of SoC and temperature. As mentioned in Section 5.1, the optimal charging power profile leads to a delayed charge that keeps lower values of SoC (as the postponed baseline) while avoiding excessive power that would cause battery overheating (as the medium baseline).

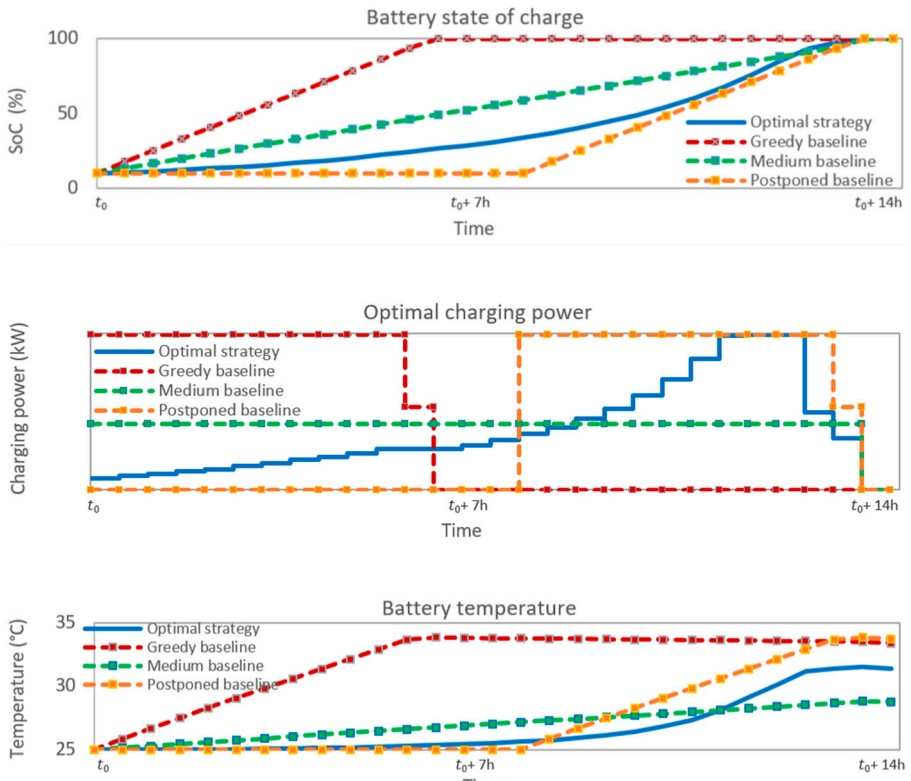

**Figure 9.** Optimal charging schedules compared with three typical charging strategies.

The simulation results in Figure 10 show that the optimal solution is the best cost effective strategy and has been able to achieve a 10% capacity fade during 10 years of operation.

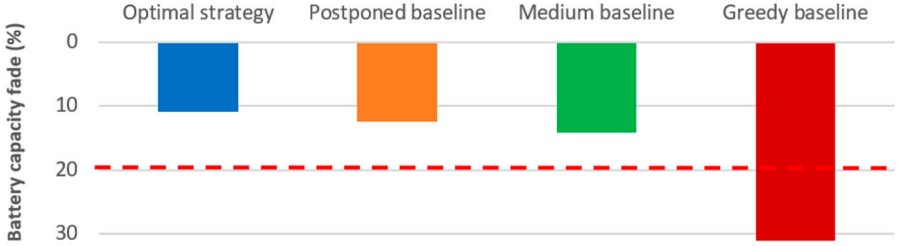

**Figure 10.** Battery capacity loss during 10 years of operation.

The postponed baseline and the medium baseline strategies cause more capacity loss but remained at a satisfactory level of 12% and 14%.

The greedy baseline strategy causes a significant capacity loss around 31%. Based on our simulation, if the bus operator uses the greedy baseline strategy, he has to replace his battery pack once during the 10 years of operation. Otherwise, if he uses the optimal strategy, he can save his battery pack during almost 20 years of operation generating a battery replacement cost saving of 155 k€ if we suppose that the battery pack costs 500 €·kWh$^{-1}$. It must be noted that the optimization results are highly dependent on the battery electro-thermal aging model.

## 6. Conclusions and Future Works

This paper introduces an intelligent charging method for electric bus fleet that minimizes the battery aging and can handle a large number of buses. A case study has been implemented to illustrate the smart charging during the night at a bus depot. The implemented optimization algorithm namely NLP achieves good performance after only 10 s of computing time for one EB, which makes it possible

to deal with cases of large fleets of several hundred buses. We have tested this approach on two EBs with different constraints in order to better understand and validate the behavior of the algorithm for a small problem. Results show a good accordance between the optimized charging strategy and the expected minimization of the criteria corresponding to battery lifetime saving while respecting the different constraints. Other strategies can be integrated into our optimization tool in future works in order to compare our strategy with more possible scenarios and optimization methods.

In future work, the purpose is to extend this method to perform a multi-objective optimization including the electricity price according to the time slot and to compare performances (in terms of minimum quality vs. computing time) with NSGA-II used in previous works. Particular attention will be paid to the battery cycling aging during the charge especially at low temperatures to study the influence of the current rate. The aging process should also take into account the bus operation during the day use.

An improvement will be made in sub-models. In particular the battery thermal model sensitivity will be carefully studied.

**Author Contributions:** A.H. performed the modeling, simulation, optimization, and wrote the original draft. All authors contributed by discussing the main ideas and participated in reviewing and editing the manuscript.

**Funding:** This work was supported by TRANSDEV.

**Conflicts of Interest:** The authors declare no conflict of interest.

## Nomenclature

| | |
|---|---|
| EVs | Electric vehicles. |
| EBs | Electric buses. |
| EVSE | Electric vehicle supply equipment. |
| V2G | Vehicle to grid |
| SoC | State of charge. |
| OCV | Open circuit voltage. |
| EOL | End of life. |
| CV | Constant voltage. |
| CC | Constant current. |
| DC | Direct current. |
| CCS | Combined charging system. |
| PLC | Power-line communication. |
| NLP | Nonlinear programming. |
| NSGA | Non-dominated Sorting Genetic Algorithm. |

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
