# Peer review of "Optimal Scheduling to Manage an Electric Bus Fleet Overnight Charging"

_energies, doi:10.3390/en12142727_

Round 1
Reviewer 1 Report
The paper is well written and offers an interesting study. However, there are several major issues which must be improved and corrected by the authors.
The similarity of the manuscript with the previously published article is relatively high (about 8%). The authors must reduce this similarity to below 4%. HIgh similarity leads to the questions related to the novelty of the article, in addition to the copyright.
There are many references which are lumped together without providing sufficient description for each of them. Please provide a short description or a descriptor for each used references, therefore, the readers can understand the content of each used references.
The paper lacks the definition of the novelty. The authors are asked to clearly explain the background of the study, including the currently available studies and the difference with them, in terms of academic findings.
Charging of bus fleet during the night leads to the large scale battery installment for the bus itself, which is considered costly and technically problematic, including the weight, etc. Could you also comment on this issue?
Charging of bus fleet in certain concentrated time leads to huge demand of electricity. Currently it is still OK to concentrate the charging during the night, however, if the charging of all EVs are concentrated during the night, the charging during the night might not be so interesting in terms of economic point of view. Therefore, distributed charging or other policy of charging should be required.
An additional literature study is demanded. Below are several works which can be used as further additional literature study, whether it relates to the charger development or charging method itself.
Battery-assisted charging system for simultaneous charging of electric vehicles. Energy 100, 82-90 (2016)
Fast-charging station deployment for battery electric bus systems considering electricity demand charges. Sustainable Cities and Society, Volume 48, July 2019, Article 101530
Electric bus fleet size and mix problem with optimization of charging infrastructure Applied Energy, Volume 211, 1 February 2018, Pages 282-295
Author Response
Point 1: The similarity of the manuscript with the previously published article is relatively high (about 8%). The authors must reduce this similarity to below 4%. High similarity leads to the questions related to the novelty of the article, in addition to the copyright.
Response 1: Concerning the high repetition rate you found in our paper, we would like to underline that the way it is computed is highly questionable. Indeed, the similarities from our own previous papers include our own names or the names of parameters as "number of simulated days". They also include text from other papers as "i=1,2,...n" or "end of the charging process" which are obvious statements or general wording for the topic treated. We have also noticed some repetitions detected from “energies” website when it is simply related to headers or foot pages of the journal template. Consequently, we completely contest the repetition rate of 28% you have attributed to our paper. Yet, we have modified the text when "detected repetition" concerned more than 2 lines. As consequence, the rate of about 8% of similarities with our previous paper is expected to be quite lower now.
Point 2: There are many references which are lumped together without providing sufficient description for each of them. Please provide a short description or a descriptor for each used references, therefore, the readers can understand the content of each used references.
Response 2: References mentioned in Section 1 [2-11] and [12-20] are detailed in Section 3. Modifications in Section 3 has been added as following:
The references [2-12] studied the centralized and decentralized overnight charging scheduling of large and small-scale EV fleet in order to minimize different objectives including V2G. The references [2;3;4;8] investigated the battery life prolongation through an optimized use of the charging/discharging process of the battery by minimizing the operational cost. References [5;6] managed a large number of EVs in order to minimize cost with respect to special grid constraints. References [7;9;10] coordinated the charging of EVs to flatten the loading profile of the transformer substation, to reduce peak demand and to minimize costs respectively while respecting EV drivers and technical constraints. In this work, we focus on the deployment of large-scale heavy vehicles (electric buses).
Point 3: The paper lacks the definition of the novelty. The authors are asked to clearly explain the background of the study, including the currently available studies and the difference with them, in terms of academic findings.
Response 3: Modifications in Section 1 have been added as following:
The mentioned studies investigated EB fleets with overnight charging on centralized bus depots. Nevertheless, they analyzed large and small-scale bus depots mostly from the perspective of minimizing the operational costs or the load peak without taking into account battery aging.
The present work differs from the existing works [13-25] as it investigates more precisely the battery electro-thermal and aging behavior in order to minimize the battery aging cost. This work differs also from our previous works [26,27] that dealt with multi-objective small-scale EBs fleet and mono-objective large-scale EBs fleet to minimize the charging cost. This work includes a large-scale EBs fleet with a possible extension of the proposed algorithm to hundreds of buses while using a mono-objective nonlinear programming optimization to minimize battery aging requiring relatively low calculation time.
To our best knowledge, there are no works concerning a centralized overnight charging scheduling of large-scale EB fleet while minimizing the battery aging.
Point 4: Charging of bus fleet during the night leads to the large scale battery installment for the bus itself, which is considered costly and technically problematic, including the weight, etc. Could you also comment on this issue?
Charging of bus fleet in certain concentrated time leads to huge demand of electricity. Currently it is still OK to concentrate the charging during the night, however, if the charging of all EVs are concentrated during the night, the charging during the night might not be so interesting in terms of economic point of view. Therefore, distributed charging or other policy of charging should be required.
Response 4: Modifications in Section 2 have been added as following:
Section 2:2.1
The charging type can be classified in two categories:
• The Overnight charging where EBs batteries are charged at the depot overnight with slow chargers (typically 40-120kW)
• The Opportunity charging where EBs batteries are charged at bus stops (up to 600 kW) or terminal (usually between 150 to 500 kW) and power charging usually through over-head pantographs.
Opportunity charging minimizes the weight of the battery, which reduces significantly the capital costs, however, the infrastructure cost remains yet very high compared to the overnight charging infrastructure. Both solutions are competitive in terms of total cost of ownership (TCO) and have their respective advantages and disadvantages. In this paper, we focus on heavy vehicles (electric buses) with overnight charging. Currently, the concentration of the charging during the night is still possible for large-scale EBs fleet. However, in a case where the electricity demand exceeds supply, the charging should be rescheduled or modulated. The proposed optimization tool can handle this issue by imposing constraints from the power grid.
Point 5: An additional literature study is demanded. Below are several works which can be used as further additional literature study, whether it relates to the charger development or charging method itself.
Response 5: The proposed references have been added as following in section 1:
Battery-assisted charging system for simultaneous charging of electric vehicles. Energy 100, 82-90 (2016): Added (Reference [11])
Fast-charging station deployment for battery electric bus systems considering electricity demand charges. Sustainable Cities and Society, Volume 48, July 2019, Article 101530: Added (Reference [22])
Electric bus fleet size and mix problem with optimization of charging infrastructure Applied Energy, Volume 211, 1 February 2018, Pages 282-295: Added (Reference [23])
Please see the attachment

Reviewer 2 Report
This paper introduces a methodical approach to manage an electric bus fleet overnight
charging. This approach identifies an optimal charging strategy in order to minimize the battery aging cost (the cost of replacing the battery spread over the battery lifetime). The work is interesting and I have the following comments:
Define LFP.
The Battery price is set at 500 €.kWh-1. A reference is required for the cost, such as from the following paper: Lai, C. S., Jia, Y., Lai, L. L., Xu, Z., McCulloch, M. D., & Wong, K. P. (2017). A comprehensive review on large-scale photovoltaic system with applications of electrical energy storage. Renewable and Sustainable Energy Reviews, 78, 439-451.
Since the cost is an important aspect and is uncertain in the economic analysis, it is ideal to examine a range of viable costs and to discuss the future implications.
“The potential economic gain of an optimal electric bus charging for 10 years operation was compared to three typical non-optimal charging strategies showing the potential economic gain.” Case studies compared with the state of the art on optimal charging strategies need to be conducted, such as:
Raab, A. F., Lauth, E., Strunz, K., & Göhlich, D. (2019). Implementation Schemes for Electric Bus Fleets at Depots with Optimized Energy Procurements in Virtual Power Plant Operations. World Electric Vehicle Journal, 10(1), 5.
Alonso, M., Amaris, H., Germain, J., & Galan, J. (2014). Optimal charging scheduling of electric vehicles in smart grids by heuristic algorithms. Energies, 7(4), 2449-2475.
The battery degradation and degradation cost aspect for lithium ion need to be enhanced in the literature review and discussion:
Lai, C. S., Jia, Y., Xu, Z., Lai, L. L., Li, X., Cao, J., & McCulloch, M. D. (2017). Levelized cost of electricity for photovoltaic/biogas power plant hybrid system with electrical energy storage degradation costs. Energy conversion and management, 153, 34-47.
Lai, C. S., Locatelli, G., Pimm, A., Tao, Y., Li, X., & Lai, L. L. (2019). A financial model for lithium-ion storage in a photovoltaic and biogas energy system. Applied Energy, 251, 113179.
Correct English: “As mentionned in Section 5.1 , the optimal charging power profile lead to a delayed charge to stay” A proof reading is required by a native English speaker, and with a correctly installed grammar and spelling software.
Include reference: According to automotive standards, 20% capacity loss generally indicates the end of life (EOL).
Author Response
Point 1: Define LFP.
Response 1: LFP has been defined as acronym for LiFePO4.
Point 2: The Battery price is set at 500 €.kWh-1. A reference is required for the cost, such as from the following paper: Lai, C. S., Jia, Y., Lai, L. L., Xu, Z., McCulloch, M. D., & Wong, K. P. (2017). A comprehensive review on large-scale photovoltaic system with applications of electrical energy storage. Renewable and Sustainable Energy Reviews, 78, 439-451.
Since the cost is an important aspect and is uncertain in the economic analysis, it is ideal to examine a range of viable costs and to discuss the future implications.
Response 2: The reference [43] has been added in Section 4 and the following text has been added:
Section 4
The battery price was set at 500 €.kWh-1. The reference [43] proposed a Li-ion battery capital cost ranges between 500-2500 $.kWh-1. Other research [45] proposed a Li-ion battery pack cost for EV ranges between 250-360 $.kWh-1 for 2015, 200-250 $.kWh-1 for 2020 and 150-200 $.kWh-1 for 2025. The cost estimation uncertainty of lithium-ion cells depends on several factors. Different values were found in the literature, thus, a value of 500 €.kWh-1 was chosen which seems more acceptable.
Section 5:5.3
Based on our simulation results, if the bus operator used the greedy baseline strategy, he has to replace his battery pack once. Otherwise, if he used the optimal strategy, he can save his battery pack during 20 years of operation generating a battery replacement cost saving of 155 k€ if we suppose that the battery pack costs 500 €.kWh-1.
Point 3: “The potential economic gain of an optimal electric bus charging for 10 years operation was compared to three typical non-optimal charging strategies showing the potential economic gain.” Case studies compared with the state of the art on optimal charging strategies need to be conducted, such as: Raab, A. F., Lauth, E., Strunz, K., & Göhlich, D. (2019). Implementation Schemes for Electric Bus Fleets at Depots with Optimized Energy Procurements in Virtual Power Plant Operations. World Electric Vehicle Journal, 10(1), 5.
Alonso, M., Amaris, H., Germain, J., & Galan, J. (2014). Optimal charging scheduling of electric vehicles in smart grids by heuristic algorithms. Energies, 7(4), 2449-2475.
Response 3: The following text has been added in Section 5:5.3
Several charging strategies have been compared to smart management strategy through the literature. Reference [7] presented an optimal scheduling strategy to minimize the load profile deviation and compared to 4 different types of EV charging strategies. Reference [44] proposed a cost-optimizing bidding strategy to minimize the operating cost. Both studies do not deal with the battery aging issue, which makes any assessments difficult to compare. Moreover, the economic gain has not been quantified. These strategies can be integrated into our optimization tool in future works in order to compare our strategy with more possible scenarios and optimization methods.
Point 4: The battery degradation and degradation cost aspect for lithium ion need to be enhanced in the literature review and discussion:
Lai, C. S., Jia, Y., Xu, Z., Lai, L. L., Li, X., Cao, J., & McCulloch, M. D. (2017). Levelized cost of electricity for photovoltaic/biogas power plant hybrid system with electrical energy storage degradation costs. Energy conversion and management, 153, 34-47.
Lai, C. S., Locatelli, G., Pimm, A., Tao, Y., Li, X., & Lai, L. L. (2019). A financial model for lithium-ion storage in a photovoltaic and biogas energy system. Applied Energy, 251, 113179.
Response 4: These references have been added as follows in section 1:
Concerning the battery aging issue in optimization, some works addressed it by developing a capacity fade model in energy system not dedicated to transport [24,25].
Point 5: Correct English: “As mentioned in Section 5.1 , the optimal charging power profile lead to a delayed charge to stay” A proof reading is required by a native English speaker, and with a correctly installed grammar and spelling software.
Response 5: In Section 5:5.1 the cited sentence has been corrected as following:
As mentioned in Section 5.1, the optimal charging power profile leads to a delayed charge remaining in lower values of SoC.
The paper has also been reviewed by a high-level English speaker.
Point 6: Include reference: According to automotive standards, 20% capacity loss generally indicates the end of life (EOL).
Response 6: A reference has been added in Section 3:3.5
The aim is to minimize the battery aging cost. According to automotive standards, 20% capacity loss generally indicates the end of life (EOL) [21]
Please see the attachment

Round 2
Reviewer 1 Report
The authors have improved sufficiently the manuscript
Author Response
The manuscript has been revised according to the academic editor comments.
Reviewer 2 Report
The paper has satisfactorily addressed my concerns.
Author Response

(The authors gave the same response as above.)
